# Use of Seashell and Limestone Fillers in Metakaolin-Based Geopolymers for Masonry Mortars

**Joseph Jean Assaad *** and **Marianne Saba**

Department of Civil & Environmental Engineering, University of Balamand, Al Kourah P.O. Box 100, Lebanon
* Correspondence: joseph.assaad@balamand.edu.lb

**Abstract:** Mortars intended for plastering and masonry works normally comply to EN 413-1 and/or ASTM C91 specifications. This paper seeks to assess the suitability of geopolymers (GPs) composed of metakaolin and seashell wastes for masonry applications. The sodium hydroxide and sodium silicate activators contained air-entraining molecules to secure about $10\% \pm 2\%$ air content. Just like the cement-based mortars, test results showed that the mechanical properties of GPs including the compressive strength, flexural strength, pull-off adhesion, and water sorptivity decreased when the seashell concentration increased in the mixture. This was mainly related to a dilution effect that reduces the aluminosilicate precursor content and formation of rigid bonds. The replacement of limestone filler by seashell powder slightly increased the mechanical properties, which was attributed to higher seashell hardness that densifies the microstructure and provides additional resistance to support the external stresses. Yet, the grinding of seashells into fine powder required higher energy than what is needed for the comminution of clinker or limestone. The use of GPs is particularly advantageous for masonry applications, as it speeds up the construction operations while eliminating the hassle of moist curing normally required with cement-based plasters.

**Keywords:** geopolymer; metakaolin; seashell; masonry; strength

## 1. Introduction

Cement manufacturing has raised serious environmental and sustainability concerns over the past few decades. Those concerns could vary from the depletion of natural raw materials and non-renewable resources to the high energy demand needed during production and the release of large amounts of greenhouse emissions per ton of finished product [1–3]. The 3Rs for waste management including reducing, reusing, and recycling are important actions to conserve natural resources, reduce the demand for energy, and minimize the negative impact on the environment [3].

Recycling is essential since improper and careless disposal of wastes leads to environmental problems. Seashells are one of those materials generated from the fishing and fish industry that result in enormous amounts of byproducts and wastes produced worldwide [4–6]. The environmental degradation caused by the dumping of seashells into the water, ocean, and along the coast has led to major environmental issues [7,8]. Seashells are hard, with a protective outer coating made by marine organisms. According to Eziefula et al. [8], seashells are majorly composed of calcium carbonate (larger than 90%), while the remainder consists of dust and contaminants. Their chemical composition is pretty comparable to that of limestone-type aggregates but contains traces of chloride and sulphate salts.

Previous research has considered the use of seashell wastes in cement-based materials for the building and construction industry [5,8,9]. Martinez-Garcia et al. [10] investigated the effect of replacing the fine and coarse aggregates by mussel shells at different ratios varying from 25% to 100% in mortar mixtures. Although the strength did not remarkably curtail with such additions, the authors noticed that the organic compounds could

detrimentally alter the aggregate–paste connection, leading to reduced hydration reactions and increased porosity. Alvarenga et al. [11] studied the fresh and hardened properties of concrete containing Peruvian scallop crushed seashell as fine aggregate; mixtures were prepared with water-to-cement ratios ($w/c$) varying from 0.75 to 0.41. The results obtained showed that the concrete performance is highly influenced by the particle size distribution and seashell replacement rate. Bamigboye et al. [4] reported that the use of organic seashells and calcareous particles instead of traditional recycled aggregates could improve the concrete's mechanical properties, while creating a more sustainable construction material. Lertwattanaruk et al. [12] attributed the increase in flexural properties during concrete testing to the high calcium content of the mussel seashell, which improves the interfacial transition zone between the cement paste and aggregates. Cuadrado-Rica et al. [13] concluded that the utilization of crushed shells as a replacement material may decrease the concrete's mechanical properties and increase the volume due to a rise of entrapped air. Uncrushed cockle shells could replace aggregate partially up to 20% with increased compressive strength, compared to non-modified concrete. Hazurina et al. [14] showed significant improvement in mechanical properties when adding seashell wastes at different percentages in concrete mixtures.

Geopolymers (GPs) are emerging materials that found wide acceptance in the building and construction industry such as for fire-resistant materials, coatings and adhesives, thermal insulation and refractory items, and waste encapsulation products [15,16]. The GPs are composed by blending the aluminosilicate precursors such as blast furnace slag, fly ash, silica fume, or metakaolin using an alkaline solution made of alkali hydroxides and silicates [17]. This reaction creates three-dimensional bond structures comprising Si-O-Al links that harden at ambient temperature, thus reducing the carbon footprint associated with the use of Portland cement.

The effect of alkaline solutions on clay minerals including metakaolin (MK) has been the subject of several studies over the years [17–19]. It was observed that the mineral phase transformation of dehydroxylated kaolinite in alkaline solutions involves two major processes including the dissolution of Si and Al species and the precipitation of hydrosodalite compounds. Chen et al. [20] reported that the mechanical properties of GP materials are directly affected by the chemical composition and fineness of the precursor as well as the type and concentration of alkaline solution. The highest strengths were achieved when the water:silicon oxide:aluminum oxide:sodium oxide:sodium hydroxide ($H_2O:SiO_2:Al_2O_3:Na_2O:NaOH$) molar ratios are equal to 11.8:3.4:1.1:0.5:1.0. De Silva et al. [21] found that the optimal reactions require balancing the aluminosilicate precursor species with the alkaline solutions made of sodium or potassium hydroxides and silicates. Hence, the fastest setting times and highest strengths occurred when the $SiO_2/Al_2O_3$ ratio varied between 2.5 and 5.01. Numerous researchers noted that the addition of increased water to compensate any loss in slump may lead to reduced compressive and flexural strengths, just like what happens when the $w/c$ increases in cement-based mixtures [22–24]. Wang et al. [25] and Cwirzen et al. [26] reported that the MK can be combined with limestone powder, given their availability at reduced cost. Cwirzen et al. [26] found that the addition of limestone increased the release of Al and Si species from MK systems activated by sodium hydroxide and silicate solutions. Qian and Song [27] found that the incorporation of 10% limestone in MK-based GP mortars led to better workability and higher mechanical properties, given the filler effect that creates compact structure.

Regarding the optimal curing conditions, it is well accepted that the geopolymerization reactions can best occur in dry conditions and relatively elevated temperatures, unlike Portland cement where moisture becomes essential for strength development [15–17]. For instance, Rovnanik [28] reported that GP specimens stored between 40 and 80 °C exhibited higher early-age strengths, as compared to companion specimens cured at ambient temperature. Similar conclusions were drawn by Kuenzel et al. [29] and Shekhovtsova et al. [30], who found that the increased curing temperature favored the strength development at early and late ages.

## 2. Context and Scope of Paper

Masonry plasters are widely used in the construction industry for plastering and rendering works [31,32]. Generally, this kind of mortar should be workable enough to facilitate the troweling and floating applications, yet with increased cohesiveness to intimately adhere to the substrate and avoid sagging or detaching problems. The fresh mortar properties (i.e., board life, water retention, and air content) are highly affected by the type and amount of the limestone fillers used to replace the Portland cement, $w/c$, chemical admixtures, and sand type and characteristics [33–35]. Like any cement-based materials, curing should be realized in wet conditions to ensure continued hydration of cement and ensure proper strength development with reduced susceptibility to shrinkage cracks [34,36]. On the hardened state, masonry plasters should withstand adverse weathering effects such as the rain, freeze–thaw cycles, excessively low or high temperature, efflorescence, and attack of aggressive sulfate or chloride ions. The use of air entrainers is recommended to enhance workability and resistance to weathering effects [32]. Typically, plasters are characterized by the pull-off bond strength, given the relevance of such property to control the internal stresses caused by thermal gradients or structural movements of the mortar–substrate composite [37,38].

The European and American standards for masonry cement (MC) distinguish three classes depending on the characteristic 28-day compressive strength that varies from 5 to 12.5 MPa, 12.5 to 22.5 MPa, and larger than 22.5 MPa [39,40]. These are, respectively, referred to as MC5, MC12.5, and MC22.5 in EN 413-1 [39], while Type N, S, and M are used by ASTM C91 [40]. The American standard specifies that the Gillmore initial and final setting times of MC pastes are not less than 90 min nor greater than 24 h, while the European standard requires those values to be 60 min and 15 h. The air content determined on mortar mixtures should vary from 8% to 22%, and water retention should be greater than 70%. The mortar proportions shall be one part MC to three parts of sand.

This paper is part of a comprehensive study undertaken to assess the feasibility of replacing the limestone by seashell materials during the manufacturing of cement-based and GP mortars intended for masonry plastering works. In fact, the increased seashell wastes estimated to about 400 billion tons per year open new opportunities for reusing such materials as the main source of calcium carbonate for producing masonry mortars [4,11]. Three classes of mortars complying to EN 413-1 and ASTM C91 requirements for masonry cement are tested in this study. The cement-based or GP-based mortars were prepared with different percentages of limestone or seashell powders. Tested properties include the liquid demand, setting times, air content, water retention, compressive/tensile strength, pull-off strength to existing substrates, and water sorptivity (or permeability). Such data can be useful to architects, engineers, and environmental activists that aim at reducing the use of Portland cement, while improving the durability and integrity of the various construction and building works.

## 3. Experimental Program

### 3.1. Materials

Portland cement complying to ASTM C150 Type I along with limestone (LF) and artificial metakaolin (MK) obtained by calcining kaolinitic clay are used in this investigation. The seashells were collected from coastal municipalities that did not have the means to dispose or recycle such wastes by sustainable means. The seashells were thoroughly washed using water to remove the dirt, salt, organic matters, and other impurities that could contaminate the final product and then dried in oven at 50 °C for 24 h.

A 22 L laboratory grinding ball mill operating at 75 rpm rotational speed was used for the grinding of the seashell materials. A total of 40 kg stainless steel balls (20 kg of 15 mm diameter and 20 kg of 5 mm diameter) were employed (Figure 1), while a glycol-based additive incorporated at 500 gr/ton was used to reduce the agglomeration phenomenon [2,41]. Grinding was stopped when the Blaine value for the seashell powders became close to the LF material. It is worth noting here that the energy required for grinding

the seashells was found to be remarkably higher than what is normally needed for the comminution of clinker or LF materials. Hence, at the required Blaine of approximately 340 m$^2$/kg, the clinker and limestone materials required, respectively, 38 and 30 min of grinding time, while this increased to 48 min for the seashells [34,41]. This reflects the increased seashell hardness and thus the need for additional energy to transform them into a fine powder. Although this may constitute a drawback from a sustainable point of view, the use of hard seashell particles led to improved mechanical properties, as will be discussed later in text.

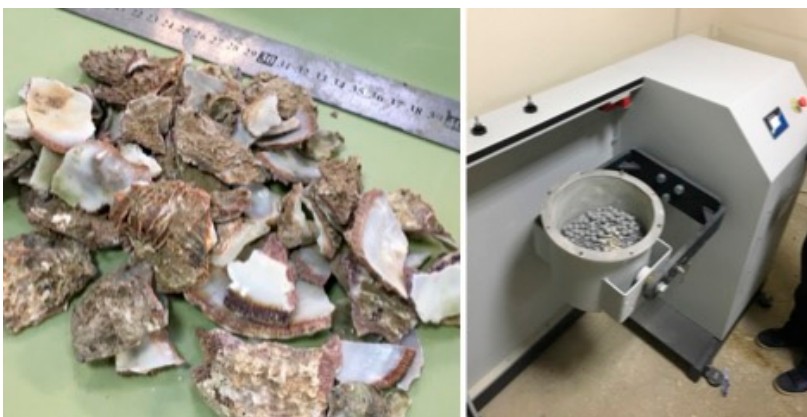

**Figure 1.** Photo of the waste seashells and grinding mill used.

The cement, LF, MK, and seashell chemical and physical properties are summarized in Table 1, while their particle size distribution curves determined using a laser diffraction analyzer are plotted in Figure 2. As shown, the MK is essentially composed of SiO$_2$ and Al$_2$O$_3$ compounds (i.e., 55% and 39%, respectively), making it excellent aluminosilicate precursor for producing GP mortars. On the contrary, the seashell materials contain about 54% calcium carbonate, reflecting its inert nature just like the limestone fillers. The MK material is extremely fine, having a B.E.T value of 19,000 m$^2$/kg, while the Blaine fineness for LF and seashell was 360 and 345 m$^2$/kg, respectively.

**Table 1.** Chemical and physical properties of cement, LF, MK, and seashell.

|  | **Cement** | **LF** | **MK** | **Seashell** |
|---|---|---|---|---|
| SiO$_2$, % | 21.4 | 5.74 | 55 | 2.27 |
| Al$_2$O$_3$, % | 4.3 | 0.15 | 39 | 0.84 |
| Fe$_2$O$_3$, % | 3.1 | 0.05 | 1.8 | 0.22 |
| CaO, % | 62.9 | 48.8 | 0.35 | 53.7 |
| MgO, % | 2.8 | 0.08 | 0.25 | - |
| SO$_3$, % | 0.45 | 1.33 | - | - |
| Na$_2$Oeq., % | 0.52 | - | - | 0.18 |
| TiO$_2$, % | - | - | 1.5 | - |
| 28-day compression, MPa | 45.5 | - | - | - |
| Blaine specific surface, m$^2$/kg | 335 | 360 | - | 345 |
| Surface area B.E.T., m$^2$/kg | - | - | 19,000 | - |
| Specific gravity | 3.15 | 2.68 | 2.2 | 2.82 |
| Loss on ignition, % | 1.85 | 41.7 | 1 | - |

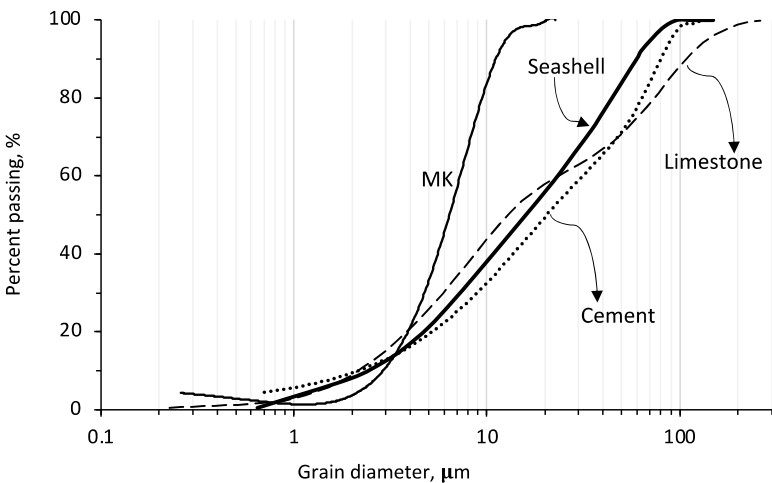

**Figure 2.** Particle size distribution curves for materials used.

A synthetic sodium dodecyl benzene sulfonate-based air-entraining agent (AEA) was used to entrain the required amount of air in the produced mortars. Its specific gravity and pH are 1.01 and 8.5, respectively. This hydrophobic surfactant works on reducing the surface tension of water, thereby leading to stabilized air bubbles that are formed during agitation [3]. Continuously graded natural sand was used for testing mortars containing masonry cement and GP binders. The sand had a bulk density, specific gravity, fineness modulus, and water absorption of 1470 kg/m$^3$, 2.63, 2.3, and 0.75%, respectively.

### 3.2. Binder Proportions and Liquid Solutions

#### 3.2.1. MC Binders

Different tests were carried out at the initial stages of this testing program in order to determine the appropriate cement replacement rates by the limestone material to meet the various strength grades required by EN 413-1 and ASTM C91 standard requirements. Hence, the cement-to-LF ratio was set on mass basis to 90%–10% for the highest strength grade MC (i.e., MC22.5-LF), while this varied to 70%–30% and 50%–50% to produce MC12.5-LF and MC5-LF binders, respectively. It is to be noted that the binder containing 70% cement was produced using the limestone or seashell materials (i.e., MC12.5-LF vs. MC12.5-Seashell) in order to compare the effect of such additions on the final product's performance. Ordinary water was used for batching; the water was premixed with 1% AEA to entrain the targeted air content in the fresh mortars, as per the EN and ASTM standards requirements.

#### 3.2.2. GP Binders

All GP binders were prepared using the MK precursor, however, while maintaining the same filler (i.e., LF and seashell) ratios determined earlier. Hence, the GP22.5-Seashell mixture contained 90% MK with 10% seashell, while those percentages varied between 70%–30% and 50%–50% for the GP12.5-Seashell and GP5-Seashell, respectively. Moreover, one GP mortar prepared with 70% MK and 30% LF was tested.

For activation, the liquid alkaline solution was prepared using sodium hydroxide (NaOH) and sodium silicate (Na$_2$SiO$_3$) solutions, while the ratio between the Na$_2$SiO$_3$-to-NaOH ratio was set at 2.0. The 10-molar (i.e., 10M) NaOH solution was prepared by mixing the NaOH beads of 98% purity with tap water at least 24 h prior to use. The sodium silicate had a specific gravity of 1.39 g/mL and SiO$_2$-to-Na$_2$O ratio of 3.25. As earlier, the AEA was added at a rate of 1% to entrain the required amount of air content, while the alkaline solution was diluted with 10% tap water to reduce the viscosity of produced GP mixtures.

The gross water-to-binder ratio (i.e., $w_{gross}$) determined by considering the total binder materials (i.e., cement, MK, LF, and seashell) as well as the effective water-to-binder ratio (i.e., $w_{eff}$) is computed as follows:

For the cement-based mixtures:

$$w_{gross} = \frac{Water}{Cement + (LF \text{ or Seashell})} \tag{1}$$

$$w_{eff} = \frac{Water}{Cement} \tag{2}$$

For the GP mixtures:

$$w_{gross} = \frac{Mass \text{ of added water} + water \text{ in } Na_2SiO_3 \text{ and } NaOH \text{ solutions}}{Mass \text{ of } MK + (LF \text{ or Seashell}) + solid \text{ content in } Na_2SiO_3 \text{ and } NaOH} \tag{3}$$

$$w_{eff} = \frac{Mass \text{ of added water} + water \text{ in } Na_2SiO_3 \text{ and } NaOH \text{ solutions}}{Mass \text{ of } MK + solid \text{ content in } Na_2SiO_3 \text{ and } NaOH} \tag{4}$$

### 3.3. Testing Methods and Procedures

The fineness and specific gravity of powder materials used in this work were measured according to ASTM C204 (Blaine method) and C188, respectively [42,43]. The water demand for MC binders to reach normal consistency was determined by mixing the binders using a measured quantity of water, as per ASTM C187 [44]. The same approach was adopted for the GP binders that were mixed with the alkaline solution. The pastes used for normal consistency were then utilized to measure the Gillmore setting times, as per ASTM C266 [45]. The various liquid demand and setting times for the MC and GP binders are summarized in Table 2. Mixing was made using a laboratory mixer, where the ambient temperature and relative humidity (RH) were 23 ± 3 °C and 55% ± 5%, respectively.

**Table 2.** Liquid demand for normal consistency and setting times for MC and GP binders.

| | Liquid Demand, mL/kg | $w_{gross}$ | $w_{eff}$ | Initial Set Time, Min | Final Set Time, Min |
|---|---|---|---|---|---|
| MC5-LF | 250 | 0.25 | 0.5 | 210 | 320 |
| MC12.5-LF | 285 | 0.285 | 0.407 | 145 | 220 |
| MC12.5-Seashell | 270 | 0.27 | 0.386 | 140 | 190 |
| MC22.5-LF | 305 | 0.305 | 0.339 | 160 | 245 |
| GP5-Seashell | 995 | 0.563 | 0.925 | 410 | 490 |
| GP12.5-LF | 1100 | 0.606 | 0.787 | 405 | 530 |
| GP12.5-Seashell | 1020 | 0.574 | 0.748 | 420 | 490 |
| GP22.5-Seashell | 1140 | 0.622 | 0.674 | 390 | 410 |

All mortar proportions consisted of one part of binder to three parts of sand on mass basis, as per EN 413-1 and ASTM C91 recommendations. The amount of water (or alkaline solution) was adjusted to secure relatively workable mortars suitable for plastering and rendering works [31,32]; a value of 160 ± 10 mm after 15 drops on the flow table was selected [34]. The mortar mixing procedure consisted of homogenizing the MC (or GP) and sand and then adding water (or alkaline solution) at low speed of 140 rpm over 2 min. After 30 s rest period, the mixing operation resumed for 1 additional minute at medium speed of 285 rpm. The ambient temperature and RH were 23 ± 3 °C and 55% ± 5%, respectively.

The air meter apparatus used to determine the air content and fresh density had 0.75 L volume, and testing was conducted as per ASTM C231 [46]. The freshly mixed mortars are subjected to controlled vacuum suction for 60 sec to assess the water retention, as per

ASTM C1506 [47]. This later property is considered as the ratio of flow determined after suction with respect to the initial value of 160 mm, multiplied by 100.

The flexural and compressive strengths ($f_r$ and $f'_c$) of hardened samples were determined using triplicate prisms having $40 \times 40 \times 160$ mm$^3$, as per EN 196-1 [48] (after flexure, the compression was determined using the two portions of broken prism). The prisms were demolded after 24 h and stored side-by-side in a closet where ambient temperature and RH were $23 \pm 3$ °C and $90\% \pm 5\%$ RH, respectively, until testing time after 7 and 28 days. Earlier studies showed that the complete immersion of GP specimens in water detrimentally alters the development of strengths [28–30].

The pull-off bond strength to existing substrates was evaluated as per BS EN 1542 [49], especially knowing the importance of such property for masonry plastering works. The $300 \times 300$ mm$^2$ concrete substrates were roughened to remove dirt and ensure proper bonding. The MC and GP mortars were applied at $5 \pm 1$ mm thickness using a masonry trowel and then allowed to cure for 28 days. The bond strength was measured using a tensile tester; the maximum applied load that causes rupture is divided by the core-drilled area having a diameter of 50 mm. Additionally, the fracture patterns including adhesive-type occurring at the interface (i.e., A-type) or mortar-type (i.e., M-type where failure takes place in the mortar itself) are noted [37,50].

The rate of water absorption (or sorptivity) is determined as per ASTM C1585 [51]. The 28-day aged specimens were oven-dried at $50 \pm 3$ °C to constant mass, prior to testing. The mortar cubes having $50 \times 50$ mm$^2$ are immersed $2 \pm 1$ mm in water, and the mass increase due to water absorption is recorded. The sorptivity is computed from the change in mass over time divided by the mortar's cross-section multiplied by water density. The mechanical properties for tested mortars containing MC and GP binders are presented in Tables 3 and 4, respectively.

**Table 3.** Physical and mechanical properties for mortars containing MC binders.

|  | **MC5-LF** | **MC12.5-LF** | **MC12.5-Seashell** | **MC22.5-LF** |
|---|---|---|---|---|
| Liquid demand, mL/kg | 258 | 264 | 250 | 290 |
| $w_{gross}$ | 1.147 | 0.838 | 0.794 | 0.716 |
| $w_{eff}$ | 0.573 | 0.587 | 0.556 | 0.644 |
| Air content, % | 12 | 10 | 13 | 12 |
| Density, kg/m$^3$ | 1920 | 1890 | 1925 | 1940 |
| Water retention, % | 88 | 86 | 81 | 74 |
| 7 d $f'_c$, MPa | 8.2 | 17.3 | 20.1 | 27.6 |
| 28 d $f'_c$, MPa | 14 | 23.5 | 26.6 | 35.4 |
| 28 d $f_r$, MPa | 1.85 | 2.65 | 2.9 | 3.4 |
| Pull-off bond, MPa | 0.31 | 0.77 | 0.76 | 0.92 |
| Sorptivity, mm/min$^{0.5}$ | 4.55 | 4.05 | 3.18 | 2.04 |

**Table 4.** Physical and mechanical properties for mortars containing GP binders.

|  | GP5-Seashell | GP12.5-LF | GP12.5-Seashell | GP22.5-Seashell |
|---|---|---|---|---|
| Liquid demand, mL/kg | 320 | 370 | 345 | 395 |
| $w_{gross}$ | 1.084 | 0.967 | 0.923 | 0.797 |
| $w_{eff}$ | 0.685 | 0.762 | 0.723 | 0.856 |
| Air content, % | 9 | 13 | 12 | 13 |
| Density, kg/m$^3$ | 1890 | 1860 | 1900 | 1910 |
| Water retention, % | 100 | 100 | 100 | 100 |
| 7 d $f_c'$, MPa | 18.7 | 32.1 | 34 | 38.3 |
| 28 d $f_c'$, MPa | 23.2 | 34.1 | 38.8 | 46.2 |
| 28 d $f_r$, MPa | 3.4 | 4.26 | 4.8 | 5.63 |
| Pull-off bond, MPa | 0.71 | 0.9 | 1.04 | 1.3 |
| Sorptivity, mm/min$^{0.5}$ | 3.2 | 2.88 | 3.01 | 1.57 |

## 4. Results and Discussion

### 4.1. Repeatability of Test Responses

Selected MC12.5-LF, GP12.5-LF, and GP12.5-Seashell mixtures were reproduced three times to assess the repeatability of test responses. The coefficient of variation (COV) was computed as the ratio between the standard deviation of various measurements and their mean value, multiplied by 100. In general, the COV values that resulted from the cement-based mortars were comparatively lower (i.e., better repeatability) than those obtained from the GP mixtures, reduced when the cement-based binder was used. Hence, for example, the COV for liquid demand to attain given consistency increased from 1.6% to 2.2% for MC12.5-LF and MC12.5-Seashell, respectively, while this varied from 6.2% to 8.8% for the compressive strength, respectively. As will be discussed later in this text, this may be attributed to the increased viscosity of the alkaline solution, thus accentuating the mixture sensitivity to flowability variations and strength development. The COV for the pull-off tests varied from 8.8% to 11% and 11.6% for the MC12.5-LF, GP12.5-LF, and GP12.5-Seashell mixtures, respectively.

### 4.2. Tests on Pastes: Normal Consistency and Setting Times

The water demand and resulting water-to-binder ratios (i.e., $w_{gross}$ and $w_{eff}$) required for normal consistency for the tested MC and GP pastes are plotted in Figure 3; the final setting times are also shown. Generally speaking, the water demand decreased for MC binders containing increased filler (i.e., LF or seashell) additions; for example, this varied from 305 to 285 and 250 mL/kg for MC22.5-LF, MC12.5-LF, and MC5-LF mixtures, respectively. The corresponding $w_{gross}$ varied from 0.305 to 0.285 and 0.25, respectively. This can be mostly attributed to refinement in the binder packing density that reduces the need of water for proper lubrication [25,34]. The substitution of LF by seashell slightly reduced the water demand (i.e., from 285 to 270 mL/kg), albeit the variations remained within the repeatability of responses. On the other hand, the longest setting times (i.e., final set of 5:20 h:min) occurred for the MC5-LF mixture, given the inert nature of the limestone materials that retards the initiation of cement hydration [25–27]. The setting gradually decreased to 3:40 and 3:10 h:min for the MC12.5 mixtures containing LF or seashell additions, respectively, due to an increased cement content in the binder.

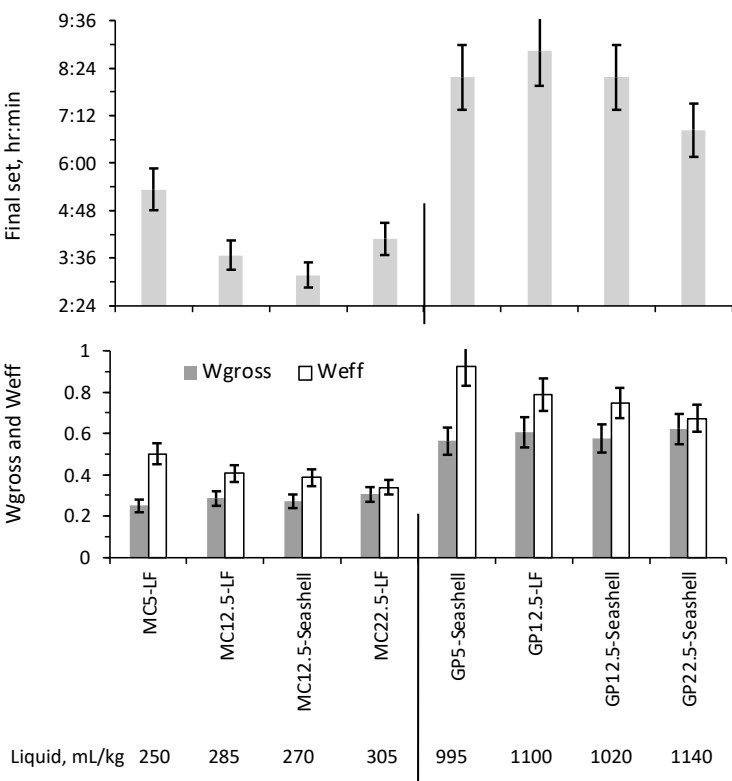

**Figure 3.** Variations in liquid demand, $w_{gross}$, $w_{eff}$, and final setting times.

The liquid solution demand necessary to achieve normal consistency for the GP binders almost tripled (i.e., varying from 995 to 1140 mL/kg), when compared to the water demand required for the MC binders (Figure 3). The corresponding $w_{gross}$ varied from 0.622 to 0.574 and 0.563 for GP22.5-Seashell, GP12.5-Seashell, and GP5-Seashell mixtures, respectively. This can be primarily attributed to the viscous nature of the alkaline solution that increases the cohesiveness of the interstitial liquid phase and hinders the ease of flowability. Palacios et al. [52] and Saba and Assaad [22] found that mixtures prepared with sodium silicate follow the Bingham model, in which the yield stress and plastic viscosity are remarkably higher than those obtained when using water. Similar findings were reported by Favier et al. [53], who found that the viscosity of the alkaline solution is 10 to 100 times higher than water, leading to increased interparticle links and colloidal interactions within the GPs. Palomo et al. [54] reported that GP pastes based on fly ash materials activated with NaOH exhibited non-Newtonian flow behavior, just like cementitious pastes. The addition of sodium silicate increased viscosity of the mixture and affected the kinetics and bonding links of hydration reactions. As shown in Figure 3, the water demand slightly decreased (i.e., from 1100 to 1020 mL/kg) when the limestone is replaced by the seashell powders in the GP12.5 mixtures.

The setting times of GP binders increased by two to three times, when compared to equivalent cement-based mixtures (Figure 3). Hence, the final setting time varied from 6:50 to 8:10 h:min for GP5-Seashell and GP22.5-Seashell binders, respectively. Earlier studies showed that the geopolymerization reactions and formation of oligomer bonds are quite complex, depending on many factors such as the NaOH molar ratios and release rate of chemical species that could alter setting times [16,17]. Chen et al. [20] reported that the shortest setting times and best strengths are achieved when the GP is prepared with metakaolin, NaOH, and sodium silicate at the ratio of 3.4:1.1:0.5.

It is to be noted that the final setting time for the GP12.5-LF binder was 8:50 h:min, i.e., relatively similar to 8:10 h:min for GP12.5-Seashell. This physically implies that the replacement of limestone by seashell does not significantly affect the geopolymerization reactions, including the development of bonds and setting times.

### 4.3. Tests on Mortars

#### 4.3.1. Water Demand

Just like the pastes, the GP mortars required a higher alkaline solution to achieve the targeted flow of 160 ± 10 mm (i.e., when compared to water used in the MC mixtures). Hence, for example, the alkaline solution varied from 320 to 395 mL/kg, while the water demand varied from 258 to 290 mL/kg (Table 3 vs. Table 4). As already explained, this can be directly related to the viscous nature of the alkaline solution that increases the mortar cohesiveness and hinders its ease of flow. The effect of replacing the limestone with the seashell powders did seem to slightly reduce the liquid demand (whether water or alkaline solution) for given flowability. Hence, the $w_{gross}$ varied from 0.587 to 0.556 for MC12.5 and from 0.762 to 0.723 for GP12.5 mixtures containing LF or seashell powders, respectively.

#### 4.3.2. Air Content and Density

Generally speaking, the air content for tested MC and GP mortars was in compliance with the EN 413-1 and ASTM C91 requirements (Tables 3 and 4); this varied within 11% ± 2.5%. It is worth noting here that the addition of the sodium dodecyl benzene sulfonate-based AEA is essential to ensure the air entrainment, especially that the preliminary tests shown that the air content in AEA-free MC and GP mortars varied from 4.5% to 6%.

The density of MC mixtures varied within limited ranges (i.e., 1920 ± 30 kg/m$^3$), while that was slightly lower (i.e., 1890 ± 30 kg/m$^3$) for the GP-based ones. For almost similar air content values, the differences in density measurements can be attributed to different specific gravities of the cement and metakaolin materials (i.e., 3.15 vs. 2.2) that could alter the weight of specimens. Additionally, the need of GPs for higher liquid demand may have contributed to reducing the specimen's weight.

#### 4.3.3. Water Retention

The water retention for all MC mortars was larger than 70%, reflecting their compliance to EN 413-1 and ASTM C91 requirements. It is to be noted that this property reflects the ability of mortars to retain mixing water that may be lost due to evaporation or suction of masonry units, thus adversely reducing board life and altering the cement hydration reactions [31,32]. Hence, the water retention increased from 74% for MC22.5-LF mortar to 86% and 88% for MC12.5-LF and MC5-LF mixtures, respectively. The improvement of water retention with increased content of limestone materials can be attributed to their increased tendency towards water absorption, thus preventing the migration of the free water under the influence of suction [25,37].

The water retention measurements remarkably improved to reach 100% for all GP mortars, regardless of the limestone or seashell additions. As already explained, this can be directly related to the viscous nature of the liquid alkaline solution that increases the mixture stickiness and resistance to water loss under the vacuuming pressure. Actually, the increased mixture stickiness was felt during the execution of the experimental program, such as the manipulation of specimens or cleaning of tools. It is worth noting here that some practitioners and engineers do not recommend excessively high levels of water retention (i.e., larger than about 95%) during masonry works, since this may reduce the interfacial bonding between the masonry blocks and applied material [32,33]. As will be discussed later, however, this was not the case in this study, as all the pull-off strengths of tested GP mixtures were all acceptable and reasonably higher than the equivalent MC mortars.

#### 4.3.4. Compressive and Flexural Strengths

Figure 4 plots the $f'_c$ and $f_r$ responses determined after 28 days for the various tested MC and GP mortars. Clearly, the strength decreases when the LF concentration increases in the cement-based mixtures; for example, the 28-day $f'_c$ varied from 35.4 MPa for MC22.5-LF containing 10% limestone to 23.5 and 14 MPa for MC12.5-LF and MC5-LF mixtures made with 30% and 50% limestone, respectively. The corresponding $f_r$ varied from 3.4 to 2.65 and 1.85 MPa, respectively. This can be directly related to a dilution effect as a result of the

non-reactive limestone nature that reduces the cement hydration reactions and formation of rigid C-S-H bonds of [22,23,25]. In the literature, it is worth noting that some researchers found that small limestone additions in the range of 5% to 10% could be beneficial to promote nucleation and hydration reactions for strength development of cementitious materials [33]. Nevertheless, the high limestone replacement rates applied in this study (i.e., 10% to 50%) appear to overshadow such phenomena and attenuate the potential associated benefits.

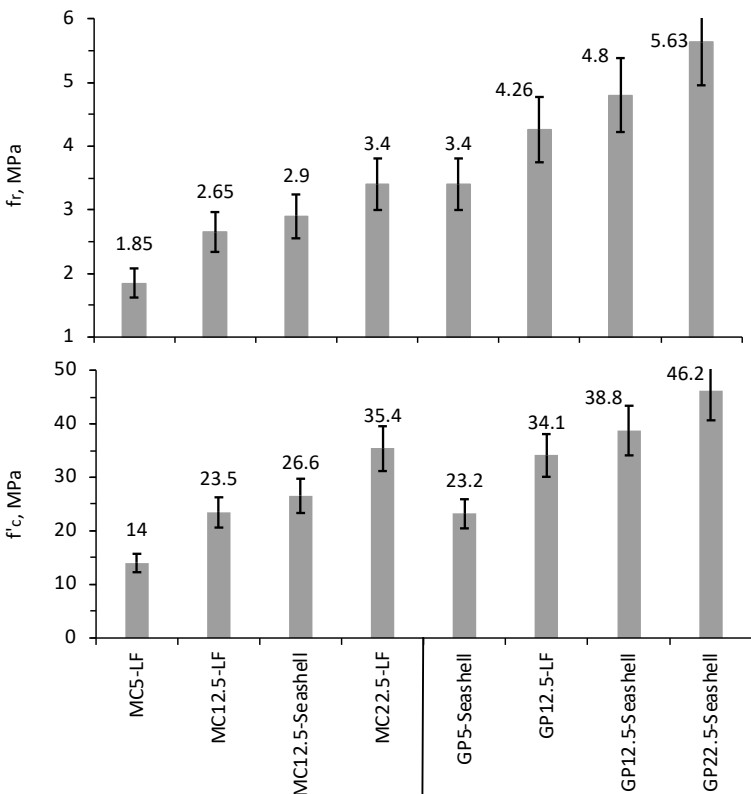

**Figure 4.** Variations in 28 d $f_c'$ and $f_r$ responses for MC and GP mortars.

The effect of replacing limestone by seashell powder slightly increased the strength properties. For example, the $f_c'$ increased from 23.5 to 26.6 MPa and $f_r$ from 2.65 to 2.9 MPa for the MC12.5-LF and MC12.5-Seashell mixtures, respectively. For almost similar Blaine fineness, the strength improvement can be attributed to increased material hardness that densifies the microstructure and provides additional resistance to support the external loading. In fact, the increased seashell hardness (compared to limestone) was noticed from the grinding tests, as the time required to grind the seashells at a given fineness of 340 m$^2$/kg was significantly longer the one needed for limestone (i.e., 48 min for the seashell vs. 30 min for limestone). In other words, it can be stated that the superior mechanical properties conferred by the seashell additions because of increased hardness could offset, or partially offset, the higher energy demand needed for their grinding.

The mechanical properties including $f_c'$ and $f_r$ of GP mixtures containing seashell powders followed similar trends as the cement-based ones. As shown in Figure 4, the higher the seashell content, the lower the strength. For example, the 28-day $f_c'$ increased from 23.2 MPa for the GP5-Seashell to 38.8 and 46.2 MPa for GP12.5-Seashell and GP22.5-Seashell mortars, respectively. As earlier, this can be mainly related to a dilution effect that reduces the aluminosilicate precursor (i.e., metakaolin) content, thus reducing the Si-O-Al rigid bonds resulting from the geopolymerization process. Additionally, the free water available in the alkaline solution increased in the GP mixtures containing higher seashell additions (i.e., w$_{eff}$ increased from 0.797 to 1.084), which may have reduced the bonds in the hardened system and contributed to reduced strength. Just like the MC mixtures, the

effect of replacing the limestone by seashell led to slightly higher strengths, which may be attributed to increased material hardness. Hence, for example, the $f'_c$ increased from 34.1 to 38.8 MPa for the GP12.5-LF and GP12.5-Seashell mortars, respectively. The corresponding $f_r$ increased from 4.26 to 4.8 MPa, respectively.

From the foregoing, it may be concluded that the waste seashells are suitable for use in GP masonry plasters complying to EN 413-1 and ASTM C91 standard requirements. Yet, it is important to highlight that the GPs cure and develop their strengths under ambient temperatures, which reduces the hassle for applying any specific wet curing conditions normally required for cement-based materials. This can be particularly relevant for masonry works to accelerate the placement operations without the need to moist cure the plastering surfaces several times during the day, for a minimum of 3–5 days. Nevertheless, particular care should be given in cold weather, since the decrease in ambient temperature (i.e., below approximately 5 to 10 °C) may alter the setting times and strength development, requiring adequate measures and procedures to be accounted for in the construction sites.

### 4.3.5. Pull-Off Strength

In the literature, the pull-off bond strengths for cementitious-based plasters applied on concrete substrates typically vary from 0.25 to 1.2 MPa. Such strengths are affected by many parameters including the surface preparation such as the removal of dirt and laitance that may deteriorate adhesion, as well as the substrate absorption properties and porosity, to achieve good interfacial bonding with the plaster [37,38]. Normally, the performance of plasters is prescribed by the 28-day compressive strength, albeit the bond strength remains more relevant to reflect the integrity and durability of the masonry application. The EN 413-1 and ASTM C91 standards do not specify any limiting value for the pull-off bond strengths.

Just like the $f'_c$ and $f_r$ variations, the pull-off strengths decreased when the limestone or seashell contents increased in the MC and GP mortars (Figure 5). For example, this varied from 0.92 MPa for the MC22.5-LF to 0.77 and 0.31 MPa for the MC12.5-LF and MC5-LF mixtures, respectively. Such variations were from 1.3 to 0.87 and 0.71 MPa for the GPs containing 10%, 30%, and 50% seashell powders, respectively. As earlier explained, this can be attributed to a dilution effect that reduces the strength development and bond formations between the mortar and substrate [23,25]. It is to be noted that failure occurred between the applied mortars and substrate, reflecting that the interface is the weakest layer in the system. On the other hand, the bond strengths of GPs were comparatively higher than those achieved from the MC mortars, for any given EN or ASTM class of products. This reflects the suitability for this kind of GPs in masonry applications requiring superior performance and bonding properties.

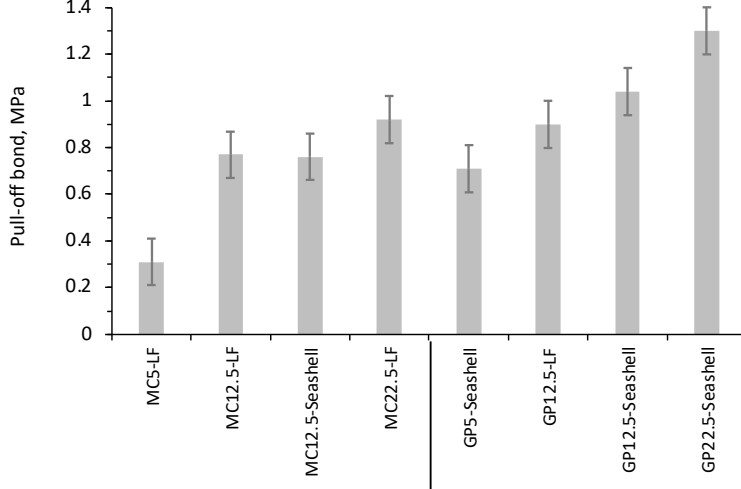

**Figure 5.** Pull-off bond strengths for MC and GP mortars.

The bonding strength did not seem to be altered with the use of limestone or seashell in the cement-based mortars; the resulting values were about 0.77 MPa. Yet, this was more significant in the GP12.5-LF and GP12.5-Seashell mortars where the bond increased from 0.9 to 1.04 MPa, respectively. Such results concur very well with the $f'_c$ and $f_r$ responses. As shown in Figure 6, adequate relationships with correlation coefficients ($R^2$) larger than 0.88 are established between the 28-day $f'_c$ with respect to $f_r$ and pull-off bond responses of various mortars.

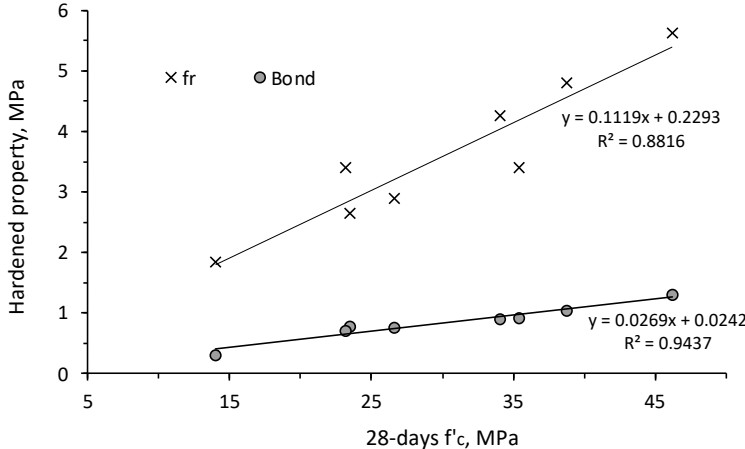

**Figure 6.** Relationships between the 28 d $f'_c$ with respect to $f_r$ and bond responses.

### 4.3.6. Sorptivity

Typical plots showing the variations of water absorption (in mm) as a function of time (in square root of minutes) for selected GP mixtures are given in Figure 7. The sorptivity is computed as the slope of the straight line fitted between the time and water absorption; the resulting $R^2$ values are quite high, reflecting that water absorption increases at a constant rate over time.

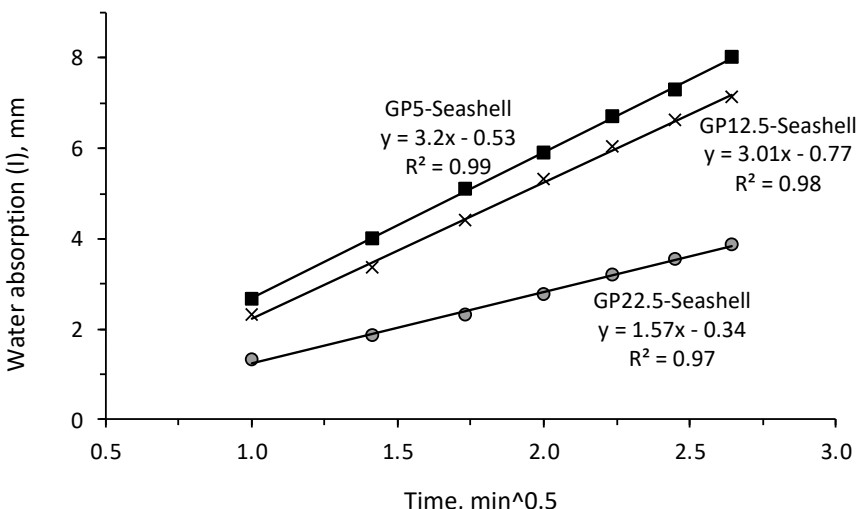

**Figure 7.** Typical plots for variations in water absorption as a function of time.

Because of the dilution effect, the effect of increasing the limestone (or, also, the seashell powder) content led to increased sorptivity measurements. For example, this increased from 2.04 mm/min$^{0.5}$ for MC22.5-LF to 4.05 and 4.55 mm/min$^{0.5}$ for MC12.5-LF and MC5-LF, respectively. Such variations were from 1.57 to 2.88 and 3.2 mm/min$^{0.5}$ for GP22.5, GP12.5, and GP5 mortars containing seashell powders. As earlier discussed, the limestone

and seashell are inert materials that could decrease the connectivity and bonds within the matrix, leading to increased capillary pores and easiness for water permeability [1,3,14].

Generally speaking, the sorptivity measurements for the GP mixtures were comparatively lower than those determined using the MC mortars. Such results are concurrent with the strength and bond properties, reflecting the suitability of GP materials for plastering works. It is to be noted that the sorptivity remarkably dropped from 4.05 to 3.18 mm/min$^{0.5}$ for MC12.5-LF and MC12.5-Seashell, respectively. Yet, the values remained within the repeatability of responses for GP mortars (i.e., 2.88 vs. 3.01 mm/min$^{0.5}$). As shown in Figure 8, a good relationship with R$^2$ of 0.8 exists between the 28-day $f_c'$ with respect to the sorptivity.

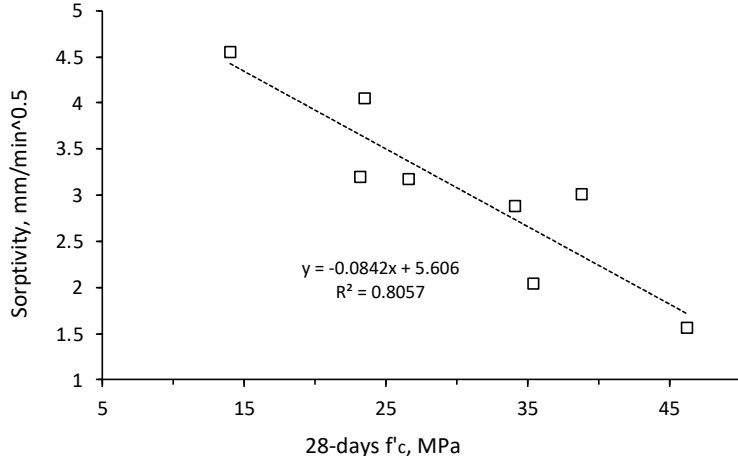

**Figure 8.** Relationship between 28 d $f_c'$ and water sorptivity.

## 5. Conclusions

Based on foregoing, the following conclusions can be warranted:

1. Because of increased hardness, the energy required for grinding the seashell into fine powder is remarkably higher than what is needed for the comminution of clinker or limestone materials. Although this may be sustainably inefficient, the superior mechanical properties conferred by the seashell additions were found to offset the higher energy demand needed for their grinding.

2. The water demand decreased for MC binders containing increased filler (i.e., LF or seashell) content, which can be attributed to refinement in the binder packing density that reduces the need of water for proper lubrication. The longest setting times occurred for the MC5-LF mixture, given the inert nature of the limestone material that retards the initiation of cement hydration.

3. The alkaline liquid solution necessary to achieve normal consistency for GP binders almost tripled when compared to the water demand required for MC binders. This was attributed to the viscous nature of the alkaline solution that increases the cohesiveness of the interstitial liquid phase and hinders the ease of flowability.

4. The replacement of limestone by seashell did not alter the setting times, whether for the cement-based or MK-based mixtures. This physically implies that such materials do not interfere with the hydration or geopolymerization reactions including the development of reacted compounds at early ages after mixing.

5. The air content for tested MC and GP mortars was in compliance with the EN 413-1 and ASTM C91 requirements. Yet, the density of MC mixtures was slightly higher than the one recorded using GP mortars, which can be attributed to different specific gravities of the cement and metakaolin materials.

6. The water retention for all GP mortars reached 100%, regardless of the limestone or seashell additions. This was directly related to the viscous nature of the liquid alkaline

solution that increases the mixture stickiness and resistance to water loss under the vacuuming pressure.

7.  Just like the cement-based mortars, the mechanical properties of GP mixtures including the compressive strength, flexural strength, pull-off bond, and water sorptivity decreased when the seashell or limestone concentration increased in the mixture. This was mainly related to a dilution effect that reduces the aluminosilicate precursor (or cement) content, thus reducing the formation of rigid bonds.

8.  The effect of replacing limestone by seashell powder slightly increased the compressive, flexural, and pull-off bond strengths. For almost similar Blaine fineness, the strength improvement can be attributed to increased material hardness that densifies the microstructure and provides additional resistance to support the external loading.

9.  The GPs do not require continuous wet curing to develop their strengths under ambient temperatures. This can be relevant for masonry works to accelerate the placement operations without the need to moist cure the plastering surfaces several times during the day.

**Author Contributions:** Conceptualization, J.J.A. and M.S.; methodology, J.J.A.; software, M.S.; validation, J.J.A. and M.S.; formal analysis, J.J.A.; investigation, J.J.A. and M.S.; resources, J.J.A.; data curation, J.J.A. and M.S.; writing—original draft preparation, J.J.A. and M.S.; writing—review and editing, J.J.A.; visualization, J.J.A.; supervision, J.J.A.; project administration, J.J.A.; funding acquisition, J.J.A. and M.S. All authors have read and agreed to the published version of the manuscript.

**Funding:** This research received no external funding.

**Data Availability Statement:** Not applicable.

**Conflicts of Interest:** The authors declare no conflict of interest.

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
