# Peer review of "Use of Seashell and Limestone Fillers in Metakaolin-Based Geopolymers for Masonry Mortars"

_minerals, doi:10.3390/min13020186_

Round 1
Reviewer 1 Report
This study investigates the use of seashell and limestone fillers in metakaolin-based geopolymers for masonry mortars. The following corrections are required for the work to be accepted.
· The use of indices in the writing of chemical oxides should be reviewed throughout the article. Exmp; Line-80, Line-84 etc.
· Fig.2 : •The unit is not shown on the “y-axis”. In addition, MK, seashell, limestone, cement expressions will be more appropriate in terms of visuality if they are given as an indicator instead of being written on a graphic.
· Please give the mixture amounts in the Table.
· Line 265-266: “Selected MC12.5-LF, GP12.5-LF, and GP12.5-Seashell mixtures were reproduced 265 three times to assess the repeatability of test responses.”Please add error bars of Figs.
· In Fig.6, the unit (Mpa) should be written on the y-axis.
· Line 466-469’da “GP22.5, GP12.5, and GP5 mortars containing seashell powders. As earlier discussed, the limestone and seashell are inert materials that could decrease the connectivity and bonds within the matrix, leading to increased capillary pores and easeness for water permeability [1,3,14].” I think there should be SEM images of the mortar matrices to confirm this. Authors should include SEM images, if available.
· In Fig 8. “y-axis” should be arranged as “Sorptivity, mm/min0.5”.
Author Response
Please, refer to attached document.

Reviewer 2 Report
Review manuscript entitled: " Use of seashell and limestone fillers in metakaolin-based geopolymers for masonry mortars"
The manuscript deals with basic properties of geopolymers (GPs) composed of metakaolin and seashell wastes for masonry applications. Three classes of mortars complying to EN 413-1 and ASTM C91 requirements for masonry cement were tested in this study. The cement-based or GP-based mortars were mixed with different percentages of limestone or seashell powders. Tested properties include the liquid demand, setting times, air content, water retention, compressive/tensile strength, pull-off strength to existing substrates, and water sorptivity (or, permeability).
The paper uses a clear scientific approach to the subject matter, which is a clear strength. Furthermore, the text is well written and concise. In the reviewer's opinion, this is a relevant work, which provides interesting findings, that deserves to be shared with the scientific community
However, improvements are required in certain important aspects of the paper, along with some minor improvements:
· First and foremost, the authors do not mention their own article entitled: “Suitability of Metakaolin-Based Geopolymers for Masonry Plastering.” , ACI Materials Journal . Nov2020, Vol. 117 Issue 6, p269-279. What are the differences between the two and what is the innovation of the submitted article?
· I suggest that you use error bars in all bar charts indicating standard deviation.
· Line 228 replace [ref]
· Line 288, replace “from the other hand” with “on the other hand”.
· Line 439, replace “from the other hand” with “on the other hand” or “contrary”.
· The authors did not observe any drying shrinkage or microcracking on the geopolymer specimens? If yes, how did they handle it?
· I would split figure 4 into two separate figures.
Final Suggestion:
The concept of the manuscript is good, and the scientific approach is solid. According to my opinion, this is a innovative work with interesting findings, that should be published.
Author Response
Please, refer to attached document.

Round 2
Reviewer 1 Report
The authors have made the necessary changes. Therefore the manuscript can be accepted.
Author Response
Thanks.
Reviewer 2 Report
The authors have adressed almost all comments.
Error bars have not been added in all figures (eg. Fig.3, Fig. 4)
Also Figure 4 is difficult to read. Either split it into 2 or avoid the bar into bar design. Consider a bar next to bar. This way you can also show the standard deviation for fr.
Also use appropriate subscripts in all Figures (fc, fr).
Author Response
Thanks very much for the comments.
As requested by the reviewers, the Fig. 3 and 4 are modified to include the error bars (also, the fig. 4 is divided into two parts).
The appropriate subscripts were used in all text and figures for f'c and fr. Thanks again.
The English writing is re-checked with a professional from our University.